# Association between the Composite Dietary Antioxidant Index and Atherosclerotic Cardiovascular Disease in Postmenopausal Women: A Cross-Sectional Study of NHANES Data, 2013–2018

**DOI:** 10.3390/antiox12091740

**Published:** 2023-09-08

**Authors:** Chenning Liu, Wenyu Lai, Meiduo Zhao, Yexuan Zhang, Yuanjia Hu

**Affiliations:** 1State Key Laboratory of Quality Research in Chinese Medicine, Institute of Chinese Medical Sciences, University of Macau, Macao SAR 999078, China; yc27555@connect.um.edu.mo (C.L.); wylai2017@i.smu.edu.cn (W.L.); mc15321@umac.mo (Y.Z.); 2Department of Public Health and Medicinal Administration, Faculty of Health Sciences, University of Macau, Macao SAR 999078, China; 3Department of Epidemiology and Biostatistics, Institute of Basic Medical Sciences Chinese Academy of Medical Sciences, School of Basic Medicine Peking Union Medical College, Beijing 100005, China; zmd@ibms.pumc.edu.cn

**Keywords:** composite dietary antioxidant index, atherosclerotic cardiovascular disease, postmenopausal women, NHANES

## Abstract

The relationship between composite dietary antioxidant index (CDAI) levels and the risk of atherosclerotic cardiovascular disease (ASCVD) in postmenopausal women is unknown. In total, 3109 women from the National Health and Nutrition Examination Survey 2013–2018 were included in this cross-sectional study. We evaluated the association between CDAI levels and the risk of ASCVD by using three logistic regression models and restricted cubic splines. A stratified analysis and sensitivity analysis were also conducted. The restricted cubic splines exhibited an L-shaped dose-response association between CDAI levels and the ASCVD risk. Logistic regression analysis found that CDAI levels were negatively associated with the occurrence of ASCVD. The ORs associated with a per-SD increase in CDAI were 0.67 (95% CI: 0.51–0.88) for ASCVD risk. Similarly, women in the group with high CDAI levels were less likely to have ASCVD (OR = 0.71, 95% CI: 0.50–0.98) compared to those in the group with low CDAI levels. When the CDAI levels were divided into quartiles, it was found that the ORs for ASCVD with CDAI levels in Q2 (−1.04–1.11), Q3 (1.11–3.72), and Q4 (3.72–43.87) were 0.63 (0.44, 0.90), 0.64 (0.42, 0.94), and 0.51 (0.27, 0.97), respectively, compared to those with CDAI levels in Q1 (−6.83–−1.04). In addition, age, high-density lipoprotein cholesterol levels, and smoking behaviors acted as potential modifiers, and ORs were more significant in women aged 40–69 years, in individuals with low high-density lipoprotein cholesterol levels, and in smokers (*p* for interaction <0.05). These findings may offer valuable insights into the role of CDAI levels in the development of ASCVD among postmenopausal women.

## 1. Introduction

Despite the lack of evidence supporting the efficacy of vitamin and mineral supplements for overall health and wellness, their consumption remains common [1]. Antioxidants, such as vitamin A or carotenoids, vitamins C and E, have been thought to reduce the risk of atherosclerotic cardiovascular disease (ASCVD) by preventing the cell damage caused by oxidative stress [2,3,4,5]. However, just last year, the United States Preventive Services Task Force (USPSTF) raised opposing views by conducting a systematic review of the efficacy and potential harms of supplementation with single nutrients, nutrient pairs, or multivitamins for reducing the risk of cardiovascular disease (CVD), cancer, and mortality in adults [6]. Based on their findings, the USPSTF recommendation opposes certain vitamin supplements, such as beta carotene and vitamin E, previously thought to have antioxidant effects, to prevent CVD [7].

The composite dietary antioxidant index (CDAI) is a summary score used to evaluate an individual’s dietary total antioxidant capacity (TAC), based on various dietary vitamins and minerals with antioxidant properties, including vitamins A, C, and E, as well as the minerals selenium and zinc [8,9]. The existing literature has indicated that CDAI is associated with specific inflammatory biomarkers, such as IL-1β and TNFα, which are known to play a role in atherosclerosis, and a high CDAI score is linked to reduced all-cause and cardiovascular mortality [10,11]. However, the relationship between CDAI and ASCVD in postmenopausal women remains unknown.

As a group, menopausal women experience a sharp increase in the incidence of ASCVD after menopause, which may be due to aging-induced oxidative stress and hormonal changes [12,13]. As life expectancy increases, with women in the United States living to around 81 years old, it is expected that approximately one-third of women’s lives will be spent postmenopause [14]. In 2021, women aged 50 and older accounted for 26% of the global female population [15], undoubtedly posing new challenges to society’s healthcare systems and warranting priority attention.

Therefore, this cross-sectional study uses data from the National Health and Nutrition Examination Survey (NHANES) 2013–2018, a large and representative survey of the US population, to investigate the potential association between CDAI and the onset of ASCVD among postmenopausal women. Our findings may offer valuable insights into the role of CDAI levels in the development of ASCVD among postmenopausal women, thus providing crucial information for the primary prevention of ASCVD in approximately one billion postmenopausal women worldwide.

## 2. Materials and Methods

### 2.1. Data Source

The NHANES is a major program carried out by the National Center for Health Statistics (NCHS) that involves the collection of nationally representative health data on the US general population. It utilizes a complex, multistage, stratified probability sampling design to collect information from approximately 5000 individuals annually through cross-sectional surveys, which have been issued on a two-year cycle since 1999. To ensure the representativeness of the sample, NHANES oversamples certain subgroups of the population, so, when conducting the data analysis, we considered sample weights to correct for differential selection probabilities, to compensate for possible inadequacies in the eligible population, and to adjust for non-coverage and non-response.

The NHANES protocol was approved by the NCHS Research Ethics Review Board, and all the participants provided informed consent. NHANES survey data, detailed survey operation manuals, consent documents, and brochures from each period are publicly available on the NHANES website (https://www.cdc.gov/nchs/nhanes, accessed on 8 February 2023).

### 2.2. Study Participants

In total, 29,400 individuals participated in the NHANES 2013–2018 cycle, for which menopausal status was determined based on self-reported menstrual information and age. The newest RHD043 item was added to the NHANES Reproductive Health Questionnaire in 2013, and it requests the reasons for the absence of menstrual periods over the past 12 months. Participants who responded “Menopause/Change of life” were classified as postmenopausal status, and women aged ≥55 years old with no menstrual information were also included in the postmenopausal group, leading to 4057 postmenopausal women in our study, in total. Participants with missing dietary (*n* = 544) or ASCVD [including coronary heart disease (CHD), angina, heart attack, and stroke, *n* = 34] information were excluded from the analysis, and covariates with missing values <10% were imputed using the random forest method. However, those with missing values ≥10%, such as the ratio of family income to poverty, were not imputed, and participants (*n* = 370) with missing information on this variable were further excluded. Therefore, our final analysis included 3109 postmenopausal women (Figure 1).

### 2.3. Outcome Ascertainment

The outcome of our research was ASCVD, defined as the presence of at least one diagnosis of CHD, angina, heart attack, or stroke according to the 2013 American College of Cardiology (ACC) and the American Heart Association (AHA) Guideline on the Treatment of Blood Cholesterol to Reduce Atherosclerotic Cardiovascular Risk in Adults [16]. Hard criteria were defined as a history of heart attack or stroke.

### 2.4. CDAI Measurement

The NHANES collected participants’ food intake data through nonconsecutive two-day 24 h-dietary recall interviews. The first dietary recall interview was conducted in person at the Mobile Examination Center (MEC), while the second was conducted over the phone 3 to 10 days later. The daily average intakes were calculated from these two days’ dietary recall data, and we computed the CDAI levels for all the subjects using a modified version developed by Wright et al. [17]. The CDAI was the sum of the daily average intakes of zinc, selenium, carotenoids, vitamin A, vitamin C, and vitamin E, which were first normalized by subtracting the mean and then dividing by their standard deviation (SD):$$CDAI=\sum_{i=1}^{n=6}(IndividualIntake-Mean)/SD$$

### 2.5. Covariates

We aimed to reduce potential confounding bias in our analysis by selecting covariates based on previous research and clinical plausibility. The selected covariates included age, race, education level, marital status, the ratio of family income to poverty, body mass index (BMI), waist circumference, alcohol use, smoking—cigarette use, moderate to vigorous recreational activities, sleep disorders, hypertension, diabetes, a family history of heart attack, neutrophil-to-lymphocyte ratio (NLR), high-density lipoprotein cholesterol (HDL-C), total cholesterol, total daily caloric intake, and total daily polyunsaturated fatty acids. Detailed measurement procedures can be found at https://www.cdc.gov/nchs/nhanes, accessed on 8 February 2023. Appendix A provides descriptions of these covariates.

### 2.6. Statistical Analysis

The categorical variables were expressed as percentages (%) and were compared using the chi-square test or Fisher’s exact test, as appropriate. Continuous data with a normal distribution were shown as means (±standard deviation [SD]) and were compared using an independent samples *t*-test. Variables with skewed distributions were presented as medians [first quantile (P25) and third quantile (P75)] and were compared using the nonparametric Wilcoxon rank sum test. We employed three logistic regression models to explore how CDAI relates to ASCVD and hard criteria in postmenopausal women. Model A did not adjust for any covariates; Model B made adjustments for age, race, education level, marital status, the ratio of family income to poverty, and BMI; Model C further adjusted for waist circumference, alcohol use, smoking—cigarette use, moderate to vigorous recreational activities, sleep disorders, hypertension, diabetes, family history of heart attack, NLR, HDL-C, total cholesterol, total daily caloric intake, and total daily polyunsaturated fatty acids based on Model B. We also assessed the presence of nonlinear dose-response relationships between CDAI levels and the risks of ASCVD, hard criteria, or components of ASCVD (CHD, angina, heart attack, and stroke), respectively, using restricted cubic splines (RCSs). We measured the interaction of the CDAI with various clinical parameters with product interaction terms. Furthermore, to investigate the modified effect of age, HDL-C levels, smoking behaviors, and CDAI on the incidence of ASCVD, we stratified the study population by age (40–59, 60–69, 70–79, and ≥80 years), HDL-C level (<50 mg/dL and ≥50 mg/dL), and smoking behaviors (smoker and non-smoker). We also conducted several sensitivity analyses to assess the robustness of our findings; in one of which, we excluded any participants who had used female hormones previously. Furthermore, in other sensitivity analyses, we separately excluded participants who were currently taking antihypertensive, lipid-lowering, or antidiabetic medication. Additionally, we excluded participants who had used statins, sodium-glucose cotransporter-2 (SGLT2) inhibitors, or angiotensin-converting enzyme (ACE) inhibitors within one month prior to their interview date. All *p* values reported were two-sided, with a significance level set at 0.05. Statistical analysis was performed using the R software (Version 4.2.2).

## 3. Results

### 3.1. Population Characteristics

The baseline characteristics of the study population with weighted estimates are presented in Table 1. In total, 3109 menopausal women were included in our analysis, of whom 453 were diagnosed with ASCVD and whose age range was 40 to 80 years old, with an average age of 64.66 ± 9.06 years. The distribution of races was as follows: 4.7% Mexican American, 4.4% other Hispanic, 75.1% non-Hispanic White, 9.2% non-Hispanic Black, and 6.7% others. The median (P25, P75) CDAI level in the ASCVD group was −1.14 (−3.05, 1.32), which was lower than that in the non-ASCVD group [−0.05 (−2.16, 2.56)]. Postmenopausal women diagnosed with ASCVD were more likely to be elderly, as well as to have a lower education level, lower income, family history of heart attack, and higher prevalence of comorbidities (hypertension, diabetes, and sleep disorders). In addition, unhealthy levels of cigarette smoking, moderate to vigorous recreational activities, waist circumference, and NLR were more prevalent among postmenopausal women in the ASCVD group.

### 3.2. Associations between CDAI Levels and Risks of ASCVD and Hard Criteria

Table 2 illustrates the associations between CDAI levels and the risks of ASCVD and hard criteria in postmenopausal women using multivariate regression models. In the crude model (Model A), it was observed that continuous CDAI levels were negatively associated with the occurrence of ASCVD and hard criteria, and the odds ratios (OR) associated with a per-SD increase in the CDAI were 0.65 (95% CI: 0.53–0.80) for ASCVD and 0.61 (95% CI: 0.50–0.75) for the hard criteria. Similarly, women in the high CDAI group were less likely to have ASCVD (OR = 0.60, 95% CI: 0.45–0.80) and hard criteria (OR = 0.57, 95% CI: 0.44–0.73) compared to the low CDAI group. When the CDAI levels were divided into quartiles, it was found that the ORs for ASCVD with CDAI levels in Q2 (−1.04–1.11), Q3 (1.11–3.72), and Q4 (3.72–43.87) were 0.63 (0.43, 0.91), 0.60 (0.41, 0.88), and 0.44 (0.26, 0.72), respectively, compared to those with CDAI levels in Q1 (−6.83–−1.04). The ORs for the hard criteria with CDAI levels in Q3 and Q4 were 0.68 (0.46, 1.00) and 0.32 (0.18, 0.54), respectively, compared to those in Q1, with a *p*-trend less than 0.001.

After adjusting for age, race, education level, marital status, the ratio of family income to poverty, and BMI (Model B), it was still found that a higher CDAI level was associated with a reduced ASCVD risk (OR = 0.71, 95% CI: 0.58–0.81, with each per-SD increase in CDAI and OR = 0.68, 95% CI: 0.51–0.90 when comparing the high CDAI level group to the low CDAI level group) and decreased hard criteria risk (OR = 0.67, 95% CI: 0.54–0.83 with each per-SD increase in CDAI and OR = 0.65, 95% CI: 0.50–0.83 when comparing the high CDAI level group to the low CDAI level group). Similarly, the ORs for ASCVD with CDAI levels in Q2, Q3, and Q4 were 0.65 (0.46, 0.93), 0.69 (0.48, 1.00), and 0.52 (0.31, 0.89), respectively, and the OR for the hard criteria with CDAI levels in Q4 was 0.38 (0.22, 0.68) compared to that with CDAI levels in Q1.

Furthermore, after additional adjustments for alcohol use, smoking behaviors, moderate to vigorous recreational activities, waist circumference, sleep disorders, hypertension, diabetes, a family history of heart attack, NLR, HDL-C, total cholesterol, total daily caloric intake, and total daily polyunsaturated fatty acids based on Model B (Model C), similar trends were observed, and it was found that a statistically lower occurrence of ASCVD and hard criteria was observed among postmenopausal women with high CDAI levels. Further, RCSs exhibited an L-shaped dose-response association between CDAI levels and the risks of ASCVD, hard criteria, or components of ASCVD (Figure 2).

### 3.3. Stratified Analyses by Potential Effect Modifiers

Stratified analyses were conducted to evaluate the association between CDAI levels and ASCVD risk in various subgroups of postmenopausal women, where high CDAI levels were consistently associated with a decreased ASCVD risk across all the subgroups. Figure 3 indicates the negative correlation between CDAI levels and ASCVD risk, and the ORs were stronger in women aged 40–69 years compared to those aged ≥70 years, in those with low HDL-C levels compared to those with high HDL-C levels, and in smokers compared to non-smokers (*p* for interaction <0.05). The ORs for ASCVD among individuals aged 40–59 years and 60–69 years were 0.41 (95% CI: 0.23–0.74) and 0.54 (95% CI: 0.31–0.93), respectively. For women aged 70–79 and ≥80 years, a per-SD increase in CDAI was associated with ORs of 0.79 (95% CI: 0.54–1.13) and 0.85 (95% CI: 0.58–1.22), respectively, for ASCVD risk. Postmenopausal women with low HDL-C levels (OR = 0.43, 95% CI: 0.27–0.72) or who smoked (OR = 0.54, 95% CI: 0.36–0.81) demonstrated lower ORs than those with high HDL-C levels (OR = 0.83, 95% CI: 0.65–1.07) or who were non-smokers (OR = 0.89, 95% CI: 0.68–1.15) when measuring the association between CDAI and the occurrence of ASCVD.

### 3.4. Sensitivity Analysis

We conducted several sensitivity analyses to verify the robustness of our results. In one sensitivity analysis, we excluded the participants who had ever used female hormones. Furthermore, in other sensitivity analyses, we separately excluded participants who were currently taking antihypertensive, lipid-lowering, or antidiabetic medication. Additionally, we excluded participants who had used statins, SGLT2 inhibitors, or ACE inhibitors within one month prior to their interview date. The sensitivity analyses demonstrated that the association between CDAI levels and ASCVD risk remained consistent with the above results (Appendix A).

## 4. Discussion

In this nationally representative cross-sectional study of the US population, we identified several significant findings. We observed a statistically lower incidence of ASCVD and hard criteria among postmenopausal women with high CDAI levels, and we found an L-shaped dose-response association between CDAI levels and the risks of ASCVD, hard criteria, or components of ASCVD. Further, our results indicated that the negative correlation between CDAI levels and ASCVD risk was stronger in women aged 40–69 years than in those aged ≥70 years, in individuals with low HDL-C levels compared to those with high HDL-C levels, and in smokers compared to non-smokers.

In recent decades, many significant risk factors for ASCVD have been identified, the prevention of which is best achieved via a healthy lifestyle. Dietary habits play a significant role in the development of obesity, hypertension, and dyslipidemia, which increase the risk of CVD [18]. Suboptimal diets are responsible for up to 45% of all cardiometabolic disease deaths, while a healthy diet, such as the Mediterranean-style diet, and antioxidants, like vitamin E, have protective effects on reducing cardiovascular morbidity and mortality [19,20,21]. As a result, guidelines have emphasized healthy diet strategies for the primary prevention of CVDs, although effective strategies are limited [22].

Despite insufficient evidence of the relationship between CDAI levels and ASCVD risk, the beneficial role of dietary antioxidants in health has been widely discussed. An increased antioxidant capacity has been linked to a reduced risk of hypertension, selected cancers, mental health issues, glucose tolerance, and central adiposity [8,9,23,24,25]. A meta-analysis of prospective cohort studies also showed that adherence to a diet with a high antioxidant capacity was associated with a decreased risk of all-cause mortality, cancer, and CVDs [26]. Further, a greater dietary TAC was associated with reduced all-cause and CVD mortality in US adults, as well as a reduced OR for certain components of metabolic syndrome, such as elevated blood pressure and diabetes [23,27,28]. Moreover, a systematic review of observational studies found that TAC is inversely related to fasting blood sugar, blood pressure, C-reactive protein, and waist circumference, and positively related to high-density lipoprotein cholesterol [29]. In addition, dietary antioxidant quality was found to predict death negatively among individuals who underwent coronary artery bypass grafting [30].

The available evidence on the relationship between dietary antioxidants and CVDs in menopausal women is limited. One observational study [31] suggested that the consumption of a diet high in TAC was inversely associated with plasma C-reactive protein levels and may contribute to CVD protection among overweight/obese postmenopausal women, which was consistent with our research. Furthermore, a cross-sectional assessment in the Kardiovize study found that dietary zinc, vitamin E, and CDAI were inversely associated with carotid intima-media thickness in women but not in men. This provided evidence that the combined intake of nutrients with antioxidant properties might prevent arterial lesions and future cardiovascular events in a sex-specific manner [32]. Two cross-sectional studies [33,34] in the adult Polish population revealed that a higher quartile DTAC was significantly linked to a lower OR for cardiovascular diseases. However, in postmenopausal Polish women, the inverse relationship between dietary polyphenol intake and CVD was observed, rather than the dietary TAC. The discrepancy observed between the general population and postmenopausal women might be related to age, sex, or other physical conditions, just as our research demonstrated that the negative correlation between CDAI and ASCVD in women aged ≥70 years was not statistically significant. Age is the major driver of CVD risk, with women aged <50 years at an almost invariably low 10-year CVD risk, while women aged >75 years are almost always at a high 10-year CVD risk [22,35,36]. Meanwhile, cigarette smoking and lipid levels are also important ASCVD risk factors [22]. The associations between CDAI and ASCVD in postmenopausal women in our research varied among different groups of different ages, HDL-C levels, and smoking behaviors, which might be influenced by these risk factors for ASCVD.

The existing literature has explored the preventive effects of vitamins on CVDs. On one hand, some studies reported that vitamins as antioxidants, vitamins A, C, and E, could contribute to protecting postmenopausal women from CVDs [37,38,39]. While carotene can be transformed into vitamin A in the body, intake via the diet was proven to reduce the risk of atherogenic dyslipidemia and acute myocardial infarction [37,40]. A study conducted in Northern Ireland indicated that vitamin C could benefit the vascular endothelia that were at risk of atherosclerosis, and there was no acute harm to vascular function [38]. Vitamin E is a widely researched natural antioxidant associated with the primary prevention of CVDs, and high intake levels via the diet could reduce the risk of CVDs [41]. Previous studies have reported that vitamin E could efficiently reduce the risk of atherosclerotic lesions caused by oxidized low-density lipoprotein (LDL) [42,43]. Moreover, a randomized controlled trial (RCT) reported that daily supplementation of vitamin E could significantly decrease non-fatal myocardial infarction and reverse arterial stiffness [44]. Despite extensive research on the role of vitamins in preventing CVDs, a systematic review by the USPSTF [7] indicated that the harms of beta carotene supplementation outweighed the benefits, and there was no net benefit of vitamin E supplementation for CVD prevention. In addition, evidence of the effect of supplementation with single, paired, or multiple vitamins on CVD mortality was inconclusive [6,45].

Zinc is a mineral that has antioxidant properties to inhibit radical production, which leads to antagonizing atherosclerotic lesions [46]. A multicenter analysis indicated that Chinese women, especially aged patients, non-smokers, and postmenopausal women, with low blood zinc concentrations were likely to develop atherosclerosis and CHD [47]. Therefore, some researchers recommended menopausal and post-menopausal women who were low in zinc intake to supplement zinc to avoid CHD [47,48]. As for selenium, despite some observational studies indicating that a low concentration correlated with an increased risk of CHD, selenium supplementation had no effect on CVD prevention [48,49,50].

To our knowledge, this is the first study to explore the relationship between CDAI and ASCVD in postmenopausal women. The large sample size and complex sampling methodology from the general population enabled us to extrapolate our findings to postmenopausal women in the United States. Furthermore, our stratified analysis identified a significant interaction between CDAI and age, smoking behaviors, or HDL-C levels, which could provide a reference for future research and diet recommendations. Moreover, our results remained robust even after accounting for a broad range of potential confounders and conducting sensitivity analysis.

However, our research has several limitations. Due to the cross-sectional nature of the survey, we were unable to determine a causal inference. In addition, while the CDAI was the sum of the daily average intakes of zinc, selenium, carotenoids, vitamin A, vitamin C, and vitamin E, we did not consider the difference between daily dietary intakes of vitamins or minerals and additional supplementation (such as from medication) of vitamins or minerals. Furthermore, although we used a nationally representative database to investigate the correlation between CDAI and ASCVD in postmenopausal women, this association cannot be directly extrapolated to the general population worldwide. Thus, additional research is required to validate our findings.

## 5. Conclusions

This cross-sectional study indicated that postmenopausal women with high CDAI levels tended to have a lower incidence of ASCVD and hard criteria, and CDAI levels had an L-shaped dose-response association with the risks of ASCVD, hard criteria, or components of ASCVD. In addition, this negative correlation was more significant among women aged 40-69, with low HDL-C levels, and who smoke.

## Figures and Tables

**Figure 1 antioxidants-12-01740-f001:**
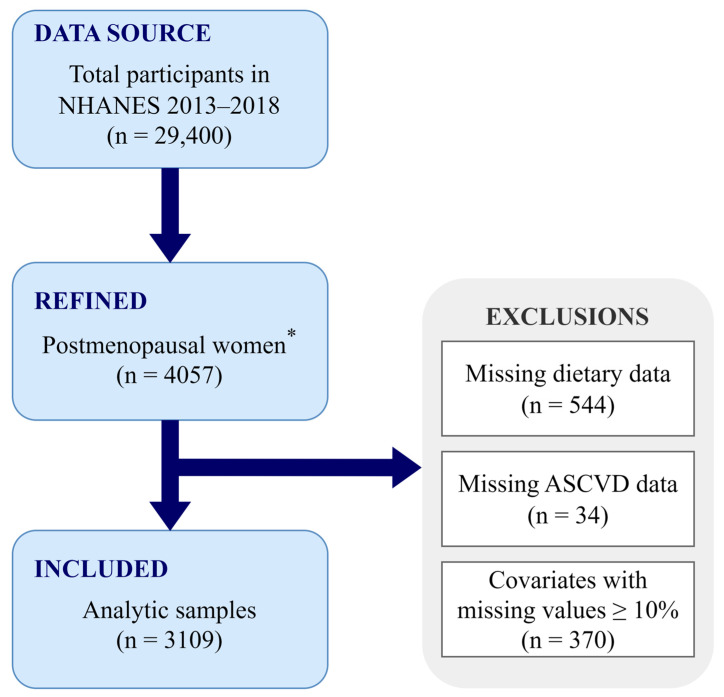
Flowchart of the study design. * Postmenopausal women were defined as participants who responded “Menopause/Change of life” or who were ≥55 years old with no menstrual information. ASCVD, atherosclerotic cardiovascular disease; NHANES, National Health and Nutrition Examination Survey.

**Figure 2 antioxidants-12-01740-f002:**
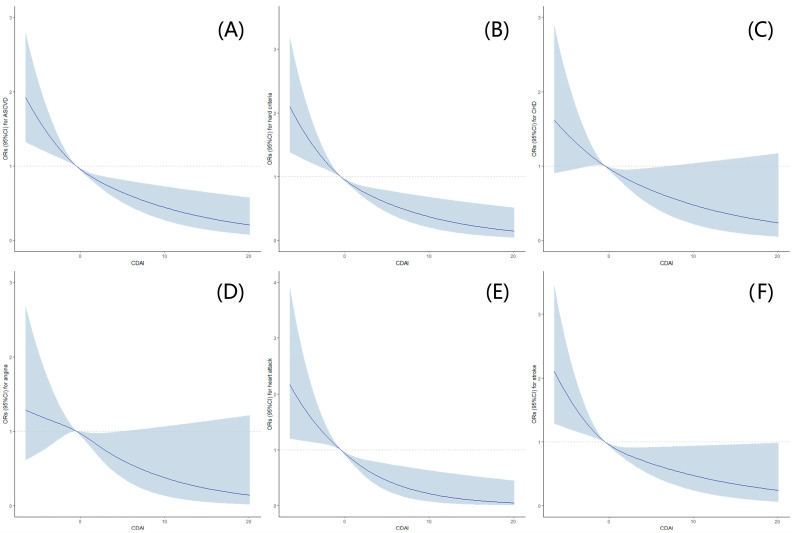
Association between CDAI Levels and the risks of ASCVD, hard criteria ^a^, or components ^b^ of ASCVD in postmenopausal women with restricted cubic splines (RCS). (**A**) Association between CDAI levels and ASCVD; (**B**) association between CDAI levels and hard criteria; (**C**) association between CDAI levels and CHD; (**D**) association between CDAI levels and angina; (**E**) association between CDAI levels and heart attack; (**F**) association between CDAI levels and stroke. The dose-response association was estimated using the nonlinear dose-response relationships between CDAI levels and the risks of ASCVD, hard criteria, or components of ASCVD. Curves represent the ORs, and blue shadings represent the 95% CI. Hard criteria ^a^ was defined as a history of heart attack or stroke. The components ^b^ of ASCVD include CHD, angina, heart attack, and stroke. ASCVD, atherosclerotic cardiovascular disease; CDAI, composite dietary antioxidant index; CHD, coronary heart disease; CI, confidence interval; OR, odds ratio.

**Figure 3 antioxidants-12-01740-f003:**
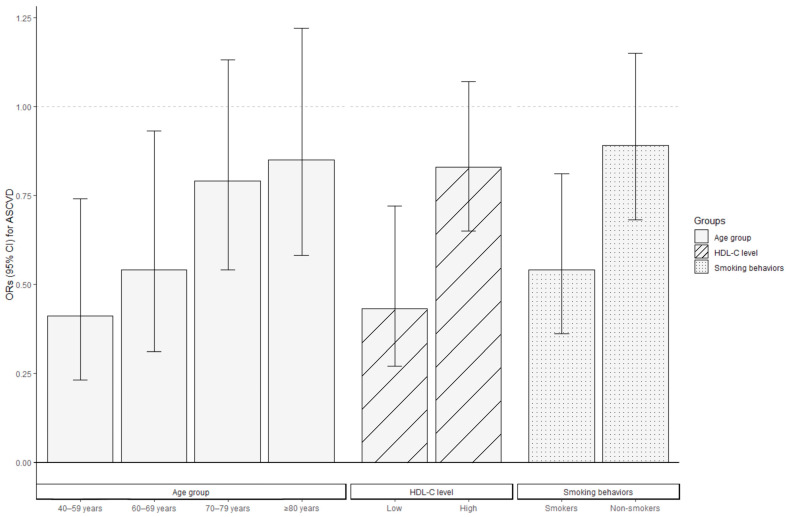
Association between CDAI levels and the ORs for ASCVD in postmenopausal women stratified by age, HDL-C level, and smoking behaviors via logistic regression. All covariates were adjusted for in the three stratified analysis models, excluding the grouping criteria for each respective group (for example, “age” was not included as a covariate in the age-stratified group). Bars represent the ORs and lines the 95% CI. ASCVD, atherosclerotic cardiovascular disease; CDAI, composite dietary antioxidant index; CI, confidence interval; HDL-C, high-density lipoprotein cholesterol; OR, odds ratio.

**Table 1 antioxidants-12-01740-t001:** Baseline Characteristics of Study Participants from NHANES 2013–2018 ^a^.

Variables	Overall (*n* = 3109)	ASCVD (*n* = 453)	Non-ASCVD (*n* = 2656)	*p*-Value
Weighted sample size	44,737,249	5,814,546	38,922,703	
Age (years)	64.66 ± 9.06	69.27 ± 8.72	63.97 ± 8.91	<0.001
Age distribution (years), *n* (%)				<0.001
40–49	80 (3.1)	3 (0.7)	77 (3.4)	
50–59	803 (30.1)	65 (15.3)	738 (32.3)	
60–69	1188 (36.1)	155 (32.1)	1033 (36.7)	
70–79	646 (20.2)	121 (28.8)	525 (19.0)	
≥80	392 (10.5)	109 (23.1)	283 (8.6)	
Race, *n* (%)				0.421
Mexican American	390 (4.7)	40 (3.2)	350 (4.9)	
Other Hispanic	365 (4.4)	52 (4.1)	313 (4.4)	
Non-Hispanic White	1359 (75.1)	234 (75.1)	1125 (75.1)	
Non-Hispanic Black	624 (9.2)	89 (10.0)	535 (9.1)	
Other Race	371 (6.7)	38 (7.6)	333 (6.5)	
Education level, *n* (%)				<0.001
Less than high school	686 (12.3)	125 (16.3)	561 (11.7)	
High school or equivalent	763 (26.3)	133 (33.5)	630 (25.2)	
College or above	1660 (61.4)	195 (50.1)	1465 (63.1)	
Marital status, *n* (%)				<0.001
Married	1470 (55.0)	165 (42.4)	1305 (56.9)	
Widowed	651 (17.9)	151 (33.4)	500 (15.6)	
Divorced	562 (17.1)	68 (12.5)	494 (17.8)	
Separated	117 (2.2)	22 (2.7)	95 (2.1)	
Never married	221 (4.9)	34 (6.2)	187 (4.7)	
Living with partner	88 (2.9)	13 (2.9)	75 (2.9)	
Ratio of family income to poverty, *n* (%)				<0.001
≤1.00	640 (12.1)	114 (15.7)	526 (11.6)	
1.01–3.00	1357 (36.1)	239 (51.0)	1118 (33.9)	
>3.00	1112 (51.8)	100 (33.3)	1012 (54.6)	
BMI (kg/m^2^)	29.81 ± 7.30	30.61 ± 7.57	29.69 ± 7.25	0.109
BMI (kg/m^2^), *n* (%)				0.125
<18.5	42 (1.6)	10 (2.3)	32 (1.4)	
18.5–24.9	1297 (25.4)	179 (21.6)	1118 (25.9)	
25.0–29.9	356 (29.7)	43 (26.1)	313 (30.2)	
≥30.0	1414 (43.4)	221 (49.9)	1193 (42.4)	
Waist circumference (cm)	99.00 [89.30, 110.50]	102.42 [92.45, 114.90]	98.50 [88.50, 109.80]	0.004
Alcohol use, *n* (%)	1457 (56.0)	189 (49.5)	1268 (57.0)	0.038
Smoking—cigarette use, *n* (%)	1208 (40.3)	227 (51.7)	981 (38.6)	<0.001
Moderate to vigorous recreational activities, *n* (%)	1179 (44.2)	124 (33.8)	1055 (45.7)	0.013
Sleep disorders, *n* (%)	1197 (42.2)	246 (56.8)	951 (40.1)	<0.001
Hypertension, *n* (%) ^b^	1827 (53.3)	365 (77.5)	1462 (49.7)	<0.001
Diabetes, *n* (%) ^c^	833 (20.2)	193 (35.9)	640 (17.9)	<0.001
A family history of heart attack, *n* (%)	488 (17.1)	119 (28.2)	369 (15.4)	<0.001
NLR	1.93 [1.48, 2.62]	2.18 [1.62, 3.02]	1.89 [1.47, 2.57]	0.001
HDL-C (mg/dL)	1.00 [0.00, 1.00]	1.00 [0.00, 1.00]	1.00 [0.00, 1.00]	0.004
Total cholesterol (mg/dL)	202.17 [176.00, 228.00]	184.00 [153.00, 216.00]	205.00 [179.00, 230.00]	<0.001
Total daily caloric intake (kcal, day), *n* (%)				0.091
<1550	1452 (42.1)	254 (49.5)	1198 (41.0)	
1550–1972	719 (25.4)	84 (24.0)	635 (25.6)	
1973–2554	580 (20.1)	78 (17.4)	502 (20.5)	
≥2555	358 (12.4)	37 (9.1)	321 (12.9)	
Total daily polyunsaturated fatty acids intake (gm)	15.02 [9.64, 22.02]	14.37 [8.68, 21.88]	15.11 [9.73, 22.04]	0.136
Female hormone use ^d^, *n* (%)	950 (30.8)	144 (33.7)	806 (30.4)	0.263
Antihypertensive medication use ^e^, *n* (%)	1596 (45.7)	330 (69.8)	1266 (42.1)	<0.001
Lipid-lowering medication use ^f^, *n* (%)	1107 (36.0)	257 (39.0)	850 (35.5)	0.304
Antidiabetic medication use ^g^, *n* (%)	625 (13.4)	149 (21.3)	476 (12.2)	<0.001
CDAI	−0.18 [−2.27, 2.46]	−1.14 [−3.05, 1.32]	−0.05 [−2.16, 2.56]	<0.001

Abbreviations: ASCVD, atherosclerotic cardiovascular disease; BMI, body mass index; CDAI, composite dietary antioxidant index; HDL-C, high-density lipoprotein cholesterol; NHANES, National Health and Nutrition Examination Survey; NLR, neutrophil-to-lymphocyte ratio. ^a^ All estimates accounted for sample weights and complex survey designs, and percentages and means were adjusted for survey weights of NHANES. ^b^ Hypertension was defined based on self-reported information, either from a doctor’s diagnosis or advice to take antihypertensive medication. ^c^ Diabetes was defined as self-reported diabetes (participants who answered “yes” to the question “Has a doctor told you that you have diabetes?”) or current use of hypoglycemic agents or insulin, or a hemoglobin A1c (HbA1c) level ≥6.5%. ^d^ Female hormone use was defined as individuals who had ever used any forms of female hormones, including pills, creams, patches, and injectables, but not for birth control or infertility purposes. ^e^ Antihypertensive, ^f^ lipid-lowering, or ^g^ antidiabetic medication use were defined as individuals currently taking them.

**Table 2 antioxidants-12-01740-t002:** Associations between CDAI Levels and the Risks of ASCVD and Hard Criteria in Postmenopausal Women ^a^.

CDAI	Model A		Model B		Model C	
	ASCVD	Hard Criteria ^b^	ASCVD	Hard Criteria ^b^	ASCVD	Hard Criteria ^b^
As continuous (per SD)	0.65 (0.53, 0.80) ***	0.61 (0.50, 0.75) ***	0.71 (0.58, 0.87) ***	0.67 (0.54, 0.83) ***	0.67 (0.51, 0.88) **	0.61 (0.46, 0.81) ***
By cut-off						
Low	Ref.	Ref.	Ref.	Ref.	Ref.	Ref.
High	0.60 (0.45, 0.80) ***	0.57 (0.44, 0.73) ***	0.68 (0.51, 0.90) **	0.65 (0.50, 0.83) **	0.71 (0.50, 0.98) *	0.65 (0.48, 0.89) **
Interquartile						
Quartile 1 (−6.83–−1.04)	Ref.	Ref.	Ref.	Ref.	Ref.	Ref.
Quartile 2 (−1.04–1.11)	0.63 (0.43, 0.91) *	0.69 (0.45, 1.06)	0.65 (0.46, 0.93) *	0.79 (0.49, 1.12)	0.63 (0.44, 0.90) *	0.70 (0.46, 1.08)
Quartile 3 (1.11–3.72)	0.60 (0.41, 0.88) **	0.68 (0.46, 1.00) *	0.69 (0.48, 1.00) *	0.74 (0.54, 1.06)	0.64 (0.42, 0.94) *	0.70 (0.46, 1.02)
Quartile 4 (3.72–43.87)	0.44 (0.26, 0.72) **	0.32 (0.18, 0.54) ***	0.52 (0.31, 0.89) *	0.38 (0.22, 0.68) **	0.51 (0.27, 0.97) *	0.33 (0.16, 0.69) **
*p*-trend	0.002	<0.001	0.022	0.002	0.042	0.005

Abbreviations: ASCVD, atherosclerotic cardiovascular disease; BMI, body mass index; CDAI, composite dietary antioxidant index; CI, confidence interval; HDL-C, high-density lipoprotein cholesterol; NLR, neutrophil-to-lymphocyte ratio; OR, odds ratio. ^a^ The associations between CDAI levels and the risks of ASCVD and hard criteria in postmenopausal women are presented as ORs (95% CI). Model A did not adjust for any covariates. In model B, adjustments were made for age, race, education level, marital status, the ratio of family income to poverty, and BMI. Model C further adjusted for waist circumference, alcohol use, smoking—cigarette use, moderate to vigorous recreational activities, sleep disorders, hypertension, diabetes, a family history of heart attack, NLR, HDL-C, total cholesterol, total daily caloric intake, and total daily polyunsaturated fatty acids based on Model B. ^b^ Hard criteria included heart attack and stroke. * *p* ≤ 0.05; ** *p* ≤ 0.01; *** *p* ≤ 0.001.

## Data Availability

Data are publicly available at https://www.cdc.gov/nchs/nhanes (accessed on 8 February 2023).

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
