# Peer review of "Association between the Composite Dietary Antioxidant Index and Atherosclerotic Cardiovascular Disease in Postmenopausal Women: A Cross-Sectional Study of NHANES Data, 2013–2018"

_antioxidants, 2023, doi:10.3390/antiox12091740_

Round 1

Reviewer 1 Report

This cross-sectional study investigated the association between the composite dietary antioxidant index and atherosclerotic cardiovascular disease in postmenopausal women using the NHANES data. The results are interesting, however, there are some major concerns:

·         The covariant of vigorous recreational activity is not sufficient, as moderate physical activity can also play a major role in CV health.

·         Polyunsaturated fatty acids should be considered, as NHANES data have shown that polyunsaturated fatty acids predicted heart disease mortality. This is important given that polyunsaturated fatty acids are antioxidants and can bias the effect of composite dietary antioxidant index.

Some minor concerns:

·         The writing needs clarity. For example, in the abstract, it mentioned “Model A”, “hard criteria”, “high CDAI group”,“low CDAI group”, and  “ORs were stronger in women”. All these expressions are not clear.

The writing needs clarity. For example, in the abstract, it mentioned “Model A”, “hard criteria”, “high CDAI group”,“low CDAI group”, and  “ORs were stronger in women”. All these expressions are not clear.

Author Response

Point-by-point reply to the comments (antioxidants-2541891)

We thank the editor and reviewers for your valuable comments and suggestions, which significantly improved the manuscript. We fully considered them and accordingly made substantial changes to the entire manuscript, particularly in the context of moderate recreational activity and polyunsaturated fatty acids in relation to the risk of atherosclerotic cardiovascular disease (ASCVD) in postmenopausal women. Furthermore, we have conducted a supplementary analysis to examine how the use of statins, sodium-glucose cotransporter-2 (SGLT2) inhibitors, or angiotensin-converting enzyme (ACE) inhibitors by the participants may impact the association between CDAI levels and ASCVD risk in postmenopausal women. Generally speaking, these updates still generate research results basically consistent with our last work. We used the revision mode in our manuscript. In the following, we will discuss in detail our responses and revisions to each of the comments from the reviewers.

Reviewer 1

Major concerns:

#1: The covariant of vigorous recreational activity is not sufficient, as moderate physical activity can also play a major role in CV health.

Thank you very much for your insightful suggestions, which significantly improved the manuscript. We have analyzed the distribution of moderate to vigorous recreational activity in ASCVD and non-ASCVD groups and have identified differences between the two groups. We additionally included moderate to vigorous recreational activity as a covariant in the logistic regression model to assess the relationship between CDAI and ASCVD in postmenopausal women. For more details, please refer to TABLE 1, TABLE2, FIGURE 3, and SUPPLEMENTARY TABLE 2. Thank you again for your valuable comments.

#2: Polyunsaturated fatty acids should be considered, as NHANES data have shown that polyunsaturated fatty acids predicted heart disease mortality. This is important given that polyunsaturated fatty acids are antioxidants and can bias the effect of composite dietary antioxidant index.

We greatly appreciate your valuable comments, which prompted us to review relevant literature. Recognizing the potential influence of daily polyunsaturated fatty acids on the relationship between CDAI and ASCVD, we have included this factor as a covariant in the logistic regression model to assess their association in postmenopausal women. For further information, please refer to TABLE 1, TABLE2, FIGURE 3, and SUPPLEMENTARY TABLE 2.

Minor concerns:

The writing needs clarity. For example, in the abstract, it mentioned “Model A”, “hard criteria”, “high CDAI group”,“low CDAI group”, and  “ORs were stronger in women”. All these expressions are not clear.

Thank you very much for pointing this out. We have carefully checked the entire manuscript to ensure that the content is expressed clearly. In addition, we have made some revisions based on suggestions from native speakers regarding inappropriate expressions, grammar errors, or spelling mistakes. The revised abstract is as follows. For more details, please refer to the Revised Manuscript.

Abstract: The relationship between composite dietary antioxidant index (CDAI) levels and the risk of atherosclerotic cardiovascular disease (ASCVD) in postmenopausal women is unknown. In total, 3,109 women from the National Health and Nutrition Examination Survey 2013-2018 were included in this cross-sectional study. We evaluated the association between CDAI levels and the risk for ASCVD by using three logistic regression models and the restricted cubic spline. Stratified analysis and sensitivity analysis were also conducted. Restricted cubic splines exhibited an L-shaped dose-response association between CDAI levels and ASCVD risk. Logistic regression analysis found that CDAI levels were negatively associated with the occurrence of ASCVD. The ORs associated with a per-SD increase in CDAI were 0.67 (95%CI: 0.51-0.88) for ASCVD risk. Similarly, women in the group of high CDAI levels were less likely to have ASCVD (OR=0.71, 95%CI: 0.50-0.98) compared to those in the group of low CDAI levels. When CDAI levels were divided into quartiles, it was found that the ORs for ASCVD with CDAI levels in Q2 (-1.04~1.11), Q3 (1.11~3.72), and Q4 (3.72~43.87) were 0.63 (0.44, 0.90), 0.64 (0.42, 0.94), and 0.51 (0.27, 0.97), respectively, compared to those with CDAI levels in Q1 (-6.83~-1.04). In addition, age, high-density lipoprotein cholesterol levels, and smoking behaviors acted as potential modifiers, and ORs were more significant in women aged 40-69 years, in individuals with low high-density lipoprotein cholesterol levels, and in smokers (p for interaction <0.05). These findings may offer valuable insights into the role of CDAI levels in the development of ASCVD among postmenopausal women.

Reviewer 2 Report

The authors have presented this manuscript as an original article. Honestly, I don't understand this definition. They thoroughly used data collected with different defined criteria by National Center for Health Statistics (NCHS). In my opinion, therefore, this manuscript may be considered a review article because the authors did not design the study, and the American Institute created different surveys!

Anyway, the manuscript should be presented better because the tables should become more usable. 

The MS should be checked by an English mother language reader. 

Author Response

Point-by-point reply to the comments (antioxidants-2541891)

We thank the editor and reviewers for your valuable comments and suggestions, which significantly improved the manuscript. We fully considered them and accordingly made substantial changes to the entire manuscript, particularly in the context of moderate recreational activity and polyunsaturated fatty acids in relation to the risk of atherosclerotic cardiovascular disease (ASCVD) in postmenopausal women. Furthermore, we have conducted a supplementary analysis to examine how the use of statins, sodium-glucose cotransporter-2 (SGLT2) inhibitors, or angiotensin-converting enzyme (ACE) inhibitors by the participants may impact the association between CDAI levels and ASCVD risk in postmenopausal women. Generally speaking, these updates still generate research results basically consistent with our last work. We used the revision mode in our manuscript. In the following, we will discuss in detail our responses and revisions to each of the comments from the reviewers.

Reviewer 2

Comments and Suggestions for Authors:

The authors have presented this manuscript as an original article. Honestly, I don't understand this definition. They thoroughly used data collected with different defined criteria by National Center for Health Statistics (NCHS). In my opinion, therefore, this manuscript may be considered a review article because the authors did not design the study, and the American Institute created different surveys!

Thank you very much for your insightful comments, which allowed us to reevaluate the nature of this paper. We carefully compared the definition and requirements of the types of publications in the journal "Antioxidants". Reviews offer a comprehensive analysis of the existing literature within a field of study. In this study, we indeed utilized data from a US database to conduct our research. However, our study involves data mining from this database to explore the association between CDAI levels and ASCVD risk in postmenopausal women. This topic and research design are original rather than a summary of existing literature. Furthermore, we thoroughly examined the use of this database for exploring other topics in the original articles published by the journal, please see [1] Larvie D Y, Armah S M. Estimated Phytate Intake Is Associated with Improved Cognitive Function in the Elderly, NHANES 2013–2014[J]. Antioxidants, 2021, 10(7): 1104. [2] Bruno R R, Rosa F C, Nahas P C, et al. Serum α-Carotene, but Not Other Antioxidants, Is Positively Associated with Muscle Strength in Older Adults: NHANES 2001–2002[J]. Antioxidants, 2022, 11(12): 2386. Thank you once again for your suggestions, which have sparked the idea of designing a cohort to collect primary data for further investigation. We greatly appreciate your insights.

Anyway, the manuscript should be presented better because the tables should become more usable. 

Thank you so much for your valuable suggestions. We have carefully checked the entire manuscript and tables, although we may not fully understand the specific issues you pointed out in the tables. Taking into consideration your comments and that of other reviewers, we have made the following modifications to the tables.

  1. Given the potential influence of moderate recreational activity and daily polyunsaturated fatty acids on the relationship between CDAI and ASCVD, we have included these two factors as covariates in the logistic regression model to assess their association in postmenopausal women. For further information, please refer to TABLE 1, TABLE2, and SUPPLEMENTARY TABLE.
  2. We additionally performed a sensitivity analysis to validate our results. We excluded participants who had used statins, SGLT2 inhibitors, or ACE inhibitors within one month prior to their interview date. This analysis demonstrated that the association between CDAI levels and ASCVD risk remained consistent with our last work. For more details, please refer to SUPPLEMENTARY TABLE 2.

Thank you again for your suggestions.

Comments on the Quality of English Language:

The MS should be checked by an English mother language reader. 

Thank you for your valuable comments. We have made some revisions based on suggestions from native speakers regarding inappropriate expressions, grammar errors, or spelling mistakes. We used the revision mode in our manuscript.

Reviewer 3 Report

Very interesting and well developed study.

Just one minor comment, and that is whether there are differences depending on whether patients take statins, SGLT2 inhibitors or ACE inhibitors, and all 3 together.

Author Response

Point-by-point reply to the comments (antioxidants-2541891)

We thank the editor and reviewers for your valuable comments and suggestions, which significantly improved the manuscript. We fully considered them and accordingly made substantial changes to the entire manuscript, particularly in the context of moderate recreational activity and polyunsaturated fatty acids in relation to the risk of atherosclerotic cardiovascular disease (ASCVD) in postmenopausal women. Furthermore, we have conducted a supplementary analysis to examine how the use of statins, sodium-glucose cotransporter-2 (SGLT2) inhibitors, or angiotensin-converting enzyme (ACE) inhibitors by the participants may impact the association between CDAI levels and ASCVD risk in postmenopausal women. Generally speaking, these updates still generate research results basically consistent with our last work. We used the revision mode in our manuscript. In the following, we will discuss in detail our responses and revisions to each of the comments from the reviewers.

Reviewer 3

Comments and Suggestions for Authors:

Very interesting and well developed study.

Just one minor comment, and that is whether there are differences depending on whether patients take statins, SGLT2 inhibitors or ACE inhibitors, and all 3 together.

Thank you for your valuable comment, which prompted us to conduct an additional analysis. We aimed to assess how the use of statins, sodium-glucose cotransporter-2 (SGLT2) inhibitors, or angiotensin-converting enzyme (ACE) inhibitors by the participants may impact the association between CDAI levels and ASCVD risk in postmenopausal women. However, we observed a low prevalence of medication use among the 3109 postmenopausal women in our study cohort, with only 245 individuals (7.88%) on statins, 5 individuals (0.16%) on SGLT2 inhibitors, and 160 individuals (5.15%) on ACE inhibitors.

Therefore, in the revised manuscript, we performed a sensitivity analysis to validate our results. We excluded participants who had used statins, SGLT2 inhibitors, or ACE inhibitors within one month prior to their interview date. This analysis demonstrated that the association between CDAI levels and ASCVD risk remained consistent with our last work. For more details, please refer to SUPPLEMENTARY TABLE 2. Thanks again for your suggestion, which provides a good idea for our future study.

Round 2

Reviewer 1 Report

Thanks for addressing my concerns.

Reviewer 2 Report

I carefully read the authors' reply. They undoubtedly improved thier manuscript but my opinion regarding its originality remain abouit the same. Anyway, if the editorial board and EiC think it my be considered an original article it is ok also for me.

No comment!

Reviewer 3 Report

The authors have properly answered my suggestions.